# Early Colonization of the Intestinal Microbiome of Neonatal Piglets Is Influenced by the Maternal Microbiome

**DOI:** 10.3390/ani13213378

**Published:** 2023-10-31

**Authors:** Jin-A Lim, Jihye Cha, Soyoung Choi, Jae-Hwan Kim, Dahye Kim

**Affiliations:** Animal Genome and Bioinformatics, National Institute of Animal Science, Rural Development Administration, Wanju-gun 55365, Republic of Korea; dlawlsdk159@korea.kr (J.-A.L.); wischa91@korea.kr (J.C.); csy7pp@korea.kr (S.C.); jkim3892@korea.kr (J.-H.K.)

**Keywords:** intestinal microbiome, neonatal piglet, microbiome colonizing

## Abstract

**Simple Summary:**

This study examined changes in piglet fecal microbiomes from birth through weaning and their associations with sows. Samples from eight sows and sixty-three piglets were analyzed, revealing that *Firmicutes, Bacteroidetes*, and *Proteobacteria* dominated early piglet microbiomes, which are crucial for intestinal balance during nursing. After weaning, piglets fed solid food had increased levels of *Christensenellaceae_R-7_group*, *Succinivibrio*, and *Prevotella*. The study also suggested that the piglet intestinal microbiome is vertically transmitted from the mother. More research is needed to understand the early establishment of piglet intestinal microbiomes by integrating factors related to sows, piglets, and the environment.

**Abstract:**

The intestinal microbiome plays a crucial role in animal health and growth by interacting with the host, inhibiting pathogenic microbial colonization, and regulating immunity. This study investigated dynamic changes in the fecal microbial composition of piglets from birth through weaning and the relationship between the piglet fecal microbiome and sows. Feces, skin, neonatal oral cavity, and vaginal samples were collected from eight sows and sixty-three piglets, and 16S genome sequencing was performed. The results revealed that *Firmicutes*, *Bacteroidetes*, and *Proteobacteria* dominated the piglet microbiome in the early stages, and *Firmicutes* and *Bacteroidetes* were crucial for maintaining a balance in the intestinal microbiome during nursing. The abundance of *Christensenellaceae_R-7_group*, *Succinivibrio*, and *Prevotella* increased in weaned piglets fed solid feed. Analysis of the microbiome from sows to piglets indicated a shift in the microbiome colonizing piglet intestines, which became a significant constituent of the piglet intestinal microbiome. This study supports the theory that the neonatal intestinal microbiome is vertically transmitted from the mother. Further research is required to integrate factors related to sows, piglets, and their environments to gain a better understanding of the early establishment of the intestinal microbiome in piglets.

## 1. Introduction

Pigs are globally recognized as one of the most important livestock species, and they are being increasingly utilized as biomedical models owing to their anatomical and physiological similarities to humans [1]. As the annual consumption of pork continues to rise, substantial efforts are being made to meet this demand through genetic selection and nutritional improvements [2,3,4]. Furthermore, in response to the prohibition of antibiotic use in animal husbandry practices in various countries, there is an ongoing quest for microbial supplements that can promote the growth and health of animals [5,6].

Microbial complexes and communities, known as microbiota, coexist in the intestines of animals [7]. The intestinal microbiome plays a crucial role in host health and growth through its interactions with the host; it inhibits pathogenic microbial colonization, produces short-chain fatty acids, and regulates immunity [8,9]. The importance of bacterial colonization of the gastrointestinal tract and its metabolic and protective functions have been reported [9]. However, whether the fetal intestine and maternal uterus harbor a microbiome prior to delivery remains controversial.

Several studies have reported that the colonization and establishment of the newborn intestinal microbiome likely begins prior to birth, as exposure to the microbiome can occur through the placenta, fetal membranes, and amniotic fluid [10,11,12]. Exposure to colonizing bacteria continues at birth and during the first year of life and profoundly influences lifelong health [13]. Establishing a beneficial microbiome is critical during the weaning stage because early colonization by beneficial bacteria in the intestinal microbiome is essential for normal intestinal and immune system function [14,15]. Moreover, it is well known that breastfeeding and exposure to the external environment affect early intestinal microbial formation during the weaning period [11,16,17,18,19]. In particular, the supplementation of fresh liquid whey in growing piglets has been confirmed to play a role in increasing beneficial bacteria and reducing harmful bacteria in the pig fecal microbiota [20,21]. As piglets are housed in the same space as their mothers until weaning, vertical transmission of the microbiome from their mothers plays an essential role in the development of the intestinal microbiome of newborns during the early growth stages [22,23]. The abrupt change in diet from breast milk to more complex nutrient sources, such as formulated feed, is a major factor influencing shifts in the piglet microbiome during weaning. Therefore, the composition of the intestinal microbiome may change with shifts in the diet of the piglets.

Notably, a portion of the early intestinal microbiome derived from sows is retained despite changes in various environmental factors, such as the hog barn, feed, water, stocking density, management, and temperature. Moreover, a balanced intestinal microbial environment is crucial for piglet growth and health [15], highlighting the importance of the early microbial community derived from sows. Therefore, it is necessary to investigate changes in the fecal microbial composition of piglets during the lactation period when piglets cohabitate with sows. The present study investigated dynamic changes in the fecal microbial composition of piglets during lactation and analyzed the relationship between the fecal microbiome of piglets and sows, and their environment from birth to weaning.

## 2. Materials and Methods

### 2.1. Animal Management and Experimental Design

Eight sows (Landrace × Yorkshire) were selected for this study, and sixty-three infant piglets (Landrace × Yorkshire × Duroc; 1.5 ± 0.3 kg average initial body weight) were cohoused with the sows by litter. Each piglet was ear notched for individual identification and dosed with a single intramuscular injection of Iron-Dextran and vaccine (circovirus and mycoplasma), following standard swine husbandry practices. The farrowing house was maintained at a temperature of 21–24 ℃ and relative humidity of 55–65%. The infant pigs were allowed to nurse freely until weaning after 21 d with creep feed. Piglets ad libitum-fed (ad libitum-fed) were allowed free access to a commercial pre-starter diet (20% crude protein, 20% crude fat, 0.35% calcium, 1% phosphorus, 1% crude fiber, 7% ash, 25,000 IU/kg vitamin A).

### 2.2. Microbial DNA Extraction and 16S rRNA Gene Sequencing

Fecal, skin, neonatal oral, milk, and vaginal samples were collected from each animal 1, 7, 14, 21, and 28 days after birth using sterile NB bio microbial transport medium (NBG-2, Noble bio, Hwaseong-si, Republic of Korea) moistened with sterile phosphate-buffered saline (pH 7.0). Fecal (rectal swabs), vaginal (vaginal swabs), skin (areolar skin swabs), neonatal oral cavity (oral swabs), and milk samples were collected from each sow, and fecal (rectal swab) samples were collected from each piglet. Prior to fecal sampling, the area around the anus was washed with sterilized PBS and collected by the rectal swab method. Neonatal oral samples were collected by swabbing the oral fluid of newborn piglets immediately after birth. Skin samples were collected by rubbing the skin of the sow’s nipple area with a cotton swab. Milk was collected by the hand milk method after washing the surrounding area with PBS prior to collection. Vaginal samples were collected by inserting the tip of the cotton swab approximately 5–8 cm into the vagina and rotating it slowly. Samples were stored at −80 °C until DNA extraction.

Microbial DNA was extracted from the samples to analyze the microbial communities in the sows and piglets. The supernatant was removed by centrifugation at 10,000× *g* for 10 min to isolate microbes in the transport medium. DNA was extracted from the collected microbes using a QIAGEN DNeasy PowerSoil Pro Kit (QIAGEN, Hilden, Germany), following the manufacturer’s instructions. The DNA concentration and purity were determined using a SpectraMax Plus 384 spectrophotometer (Molecular Devices, San Jose, CA, USA). First, we used stool samples from piglets to characterize the fecal microbiota of the piglets and then used samples from sows to assess the bacterial composition vertically transmitted from sows to piglets. The extracted DNA was stored at −80 °C until subsequent analysis.

The extracted DNA was amplified using primers (341F/806R) specific to the hypervariable regions V3–V4 of the 16S rRNA gene. The amplification program consisted of 1 cycle at 95 °C for 3 min, followed by 25 cycles at 95 °C for 30 s, 55 °C for 30 s, and 72 °C for 30 s, and a final step at 72 °C for 5 min. The PCR products were purified using AMPure XP beads (Beckman Coulter, Brea, CA, USA) to remove free primers and primer dimers. A Nextera XT Index kit (Illumina, San Diego, CA, USA) was used to attach the index sequences to the purified PCR products and create libraries. DNA quality and product size were measured using a DNA 7500 chip and Bioanalyzer 2100 (Agilent Technologies, Santa Clara, CA, USA). The DNA libraries were sequenced using an Illumina MiSeq platform.

### 2.3. Microbial Data Analysis

Microbial community analysis was performed using QIIME2 software (Quantitative Insights into Microbial Ecology, version 2020.8) [24]. The DADA2 program was used to remove chimeric reads and low-quality sequences (quality score < 25) to obtain high-quality sequences, and amplicon sequence variants (ASVs) were classified using the dada2-denoise-paired method [25]. ASVs were classified based on the Silva 138SSU reference database, and a phylogenetic tree was constructed using a fast-tree algorithm [26]. The Jvenn program was used to visualize the number of communal and differential microbiomes between the different growth periods of the piglets [27]. The alpha diversity of the microbial community was calculated using the Shannon diversity index and observed features to evaluate microbial diversity and richness. Principal coordinate analysis (PCoA) was conducted using the Bray–Curtis distance to measure the distances between the microbial compositions of the samples [28]. The microbial diversity results were visualized using the phyloseq package (v 1.40.0) in R [29].

### 2.4. Statistical Analysis

Wilcoxon tests were used to calculate *p*-values [30]. The Bayesian community-wide microbial source tracking algorithm, SourceTracker, was used to estimate the contribution of sow influence on piglet intestinal microbiome formation [31]. The algorithm determines the proportion of each “source” (sow fecal, skin, neonatal oral cavity, and vaginal samples) that contributes to the “sink” (piglet fecal sample).

## 3. Results

### 3.1. 16S rRNA Gene Sequencing for Species and Strain-Level Microbiome Analysis

To investigate the microorganisms of maternal origin involved in the formation of piglet intestinal microbiota, we collected fecal and oral samples from sixty-two healthy piglets as well as fecal, vaginal, skin, and breast milk samples from eight sows. Fecal samples were collected from piglets 1, 7, 14, 21, and 28 days after birth (Figure 1a). A total of 355 samples were processed to extract microbial DNA, and sequence information from the V3–V4 region of the 16S rRNA gene was generated for subsequent analysis. A total of paired-end reads of 400bp were acquired. A total of 4,784,260, 4,643,977, 4,062,590, 3,708,105, 3,553,611, and 3,653,236 raw reads each were generated for piglet samples on days 1, 7, 14, 21, and 28, and sow samples, respectively (Appendix A). Using the DADA2 program, we obtained a total of 20,297,621 high-quality reads. Additionally, by using the Silva 138SSU reference with a 97% species similarity threshold, we obtained 9685, 2392, 2998, 4031, 5756, and 7480 operational taxonomic units (OTUs) for piglet samples on days 1, 7, 14, 21, and 28 days and sow samples, respectively.

### 3.2. Composition of the Gut Microbiome in Piglets

To characterize the intestinal microbiome of piglets during the early growth period, we compared the relative abundances of major bacterial phyla. The dominant phyla observed in piglets during all growth periods were *Firmicutes*, *Bacteroidetes*, and *Proteobacteria*. The relative abundance of *Firmicutes* in the microbiome of piglets was high at approximately 79% immediately after birth (day 1) but decreased to approximately 57% by day 28. In contrast, *Bacteroidetes* had a low abundance of approximately 5% in the fecal samples, but gradually increased from day 7 onwards and reached approximately 25% by day 28. The relative abundance of *Proteobacteria* was approximately 9% immediately after birth, increased to approximately 20% on day 7, and then gradually decreased (Figure 1c).

The core bacterial populations present in all piglet period groups included *Clostridium sensu stricto 1*, *Turicibacter*, *Romboutsia, Streptococcus*, and *Bacteroidetes*. These were identified as significant constituents of the intestinal microbiome, with a relative abundance of >1% (Appendix A, Figure 1d). The relative abundance of *Streptococcus*, *Lactobacillus*, *Escherichia-Shigella*, and *Bacteroidetes* increased on day 7, whereas that of *Succinivibrio*, *Christensenellaceae R-7 group*, and *Prevotella* increased on day 21 (Figure 1b).

### 3.3. Microbial Diversity in Piglets from Birth to Weaning

We compared the microbial abundance and diversity in the intestinal microbiome of piglets from birth to weaning. Alpha diversity was the highest on day 1 and the lowest on day 7 in the feces, but gradually increased with age. Notably, the piglets showed higher microbial diversity on days 1 and 28 than that of the fecal microbiome of the sows (*p* < 0.05, Figure 2a). To examine the similarity of the microbial communities between piglets and sows, we performed principal coordinate analysis (PCoA) based on the Bray–Curtis distance. Our results showed that the microbial composition in the intestines of piglets changed over time, with the microbial community on day 1 showing an apparent difference from that observed in other periods (7, 14, 21, and 28 d after birth). Furthermore, on day 1, the microbial community of the piglets was more similar to that of the sows than to the microbial community observed at the other periods (Figure 2b).

### 3.4. Transmission of the Maternal Microbiome from Sows to Piglets

To investigate the effects of vertical transmission of the maternal microbiome on intestinal microbiome formation in piglets, the analysis was conducted on piglets at two stages: those born with a significant influence of the maternal microbiome on day 1, and those that were substantially affected by environmental factors on day 28. SourceTracker analysis was used to determine the proportion of the microbial communities derived from maternal sources at each stage.

The results showed that on day 1, the piglets had a higher proportion of the maternal microbiome (average 80%) than on day 28 (average 37%). Specifically, colostrum (34%), skin (26%), and neonatal oral cavity (10%), representing maternal sources, significantly decreased to 10, 7, and 3%, respectively, on day 28. However, maternal feces, which accounted for 6% of the piglet microbiome on day 1, increased to 10% by day 28 (Figure 3a). Furthermore, on day 28, the piglets had a high proportion of unknown sources, suggesting that the external environment influenced their intestinal microbiome. To identify the microbiome transferred from sows, shared operational taxonomic units (OTUs) between piglets and their mothers were compared. On day 1, piglets shared 2137, 1936, 1487, 1289, and 818 OTUs in the colostrum, skin, neonatal oral cavity, vagina, and feces with their mothers, respectively. On day 28, piglets shared 920, 887, 748, 748, and 531 OTUs with their mothers in the colostrum, skin, neonatal oral cavity, vagina, and feces, respectively (Figure 3b). Consistent with the results of the SourceTracker analysis, these results suggest that the maternal colostrum had the most significant effect on the piglet intestinal microbiome.

We compared the 16S rRNA sequences of sows and piglets to investigate the microbial transfer from sows to piglets. This analysis revealed that the microbes that differed from those of the mothers at birth mainly consisted of *Actinobacteria*, *Firmicutes*, *Bacteroidetes*, and *Proteobacteria*. Although the transferred microbiome decreased as the piglets grew, it persisted within their intestines even when they were separated from the sows. The transferred microbiome included significant piglet intestinal microbiota-forming microbes, such as *Muribaculaceae, Bacteroidetes*, *Christensenellaceae_R-7_group*, *Prevotella,* and *Lactobacillus* (relative abundance > 1%; Figure 4, Appendix A).

## 4. Discussion

The role of the intestinal microbiome in maintaining host health is multifaceted, encompassing the breakdown of indigestible fibers and the production of essential amino acids and vitamins, as previously highlighted [32,33,34,35,36]. Numerous factors play a role in shaping the initial composition of the intestinal microbiome; these factors include the method of nutrient provision, the circumstances surrounding birth, and the transmission of microorganisms from the mother [37,38]. In newborns, the biomass of the microbial community is considerably low, rendering them dependent on external microbiota to perform crucial physiological functions and maintain a balanced intestinal environment [38]. Given that the immune system of newborns is not yet fully developed, the intestinal microbiome plays a pivotal role in supporting and safeguarding them against pathogenic infections and diseases [39,40]. Furthermore, the presence of an appropriate intestinal microbiota in newborns improves nutrient absorption, curbs the colonization of harmful microorganisms, and encourages the development of the intestinal immune system [41,42,43]. Therefore, intestinal colonization by beneficial microbiomes is essential in the early stages of life. The present study aimed to investigate changes in the intestinal microbial community of piglets from birth through weaning and to elucidate the influence of the maternal microbiome on the formation of the intestinal microbiome of the piglet.

Shortly after birth, piglet microbiomes undergo rapid changes due to exposure to the external environment. The present study found that in the early stages, the piglet microbiome was primarily dominated by *Firmicutes*, *Bacteroidetes*, and *Proteobacteria*. *Firmicutes* play a pivotal role in maintaining a balanced intestinal microbiome in newborns [44]. Among these, *Lactobacillus*, which is predominantly classified as a beneficial bacterium, exhibits various important physiological functions, such as aiding in digestion and supporting the immune system [45,46]. *Bacteroidetes* are also dominant in the intestinal microbiome of newborns and utilize lactose in breast milk as an energy source [47,48]. The abundance of *Bacteroidetes* increased from days 1 to 7 in piglets fed only breast milk and dramatically decreased from days 14 to 28 during the weaning period. Additionally, the abundances of *Streptococcus*, *Lactobacillus*, *Escherichia-Shigella*, and *Bacteroidetes* increased on day 7, which was consistent with the findings of previous studies [49,50,51]. *Escherichia-Shigella* is a microbe that causes diarrhea and indigestion in newborns and colonizes the piglet intestine through the mucus in the birth canal [52,53,54]. Breast milk can inhibit the colonization of *Escherichia-Shigella* in piglets by supplying immune cells, promoting the growth of beneficial bacteria and inhibiting pathogen–intestine binding [55]. *Streptococcus* and *Lactobacillus* are the dominant bacteria found in sow milk and are transferred to piglets via suckling [22,56]. In the present study, in piglets fed solid feed on day 21, the abundance of *Christensenellaceae_R-7_group*, *Succinivibrio*, and *Prevotella* increased, which is geared towards the utilization of plant-based carbohydrates. *Succinivibrio* and *Prevotella* utilize complex carbohydrates to generate short-chain fatty acids, playing a crucial role in feed efficiency [57].

The findings of this study revealed that the piglets had a highly diverse intestinal microbiome from day 1 onward. Notably, these findings are consistent with those of Collado et al. [10] and Aagaard et al. [11], who suggested that the colonization and establishment of the intestinal microbiome of newborns commence prior to birth. This initiation is attributed to the transmission of the maternal microbiome to the fetus through the placenta, fetal membranes, and amniotic fluid. A growing body of evidence now suggests that this seeding process may even commence during fetal development. Recent studies have focused on the maternal gut as a potential source of the fetal gastrointestinal microbiota [58]. In particular, Schubbert et al. have suggested that orally ingested microbial DNA can be transferred to the fetus in mice, suggesting the presence of a trans-placental route for bacterial transfer [59]. This emerging understanding challenges the traditional view and highlights the importance of maternal gut microbiome contributions to the early neonatal microbiota.

In the present study, the diversity of the intestinal microbiome in piglets decreased rapidly within the first week after birth but gradually increased until four weeks. These results suggest that various microbes were introduced into the intestines of neonatal piglets and only host-specific microbes were retained. In addition, the composition of the intestinal microbiome became uniform as piglets grew. Because piglets cohabitate with sows before weaning, they are greatly influenced by the maternal microbiome. Thus, the microbial composition in piglet feces on day 1 was closely related to the microbiome of the mother’s milk, skin, and amniotic fluid, indicating that piglets are mainly influenced by the maternal microbiome during delivery. On day 1, the neonatal piglets consumed sow milk, suggesting that breast milk influenced the composition of the gut microbiome in piglets. The placenta produces amniotic fluid during fetal growth, which is vital for fetal health and development. The interactions among the microbiome, neonatal intestine, and amniotic fluid may affect the growth of the intestinal microbiome, along with the nutrients provided to the fetus through the placenta [11,60]. Our results also confirmed that the microbial compositions in the feces and amniotic fluid on day 1 were closely related. This finding is consistent with those of previous studies, which reported that the microbiome detected in the feces of piglets on day 1 was the same as that in the amniotic fluid to which piglets were exposed in utero [61].

In addition, our analysis of the impact of the sow-derived microbiome on piglet feces revealed that breast milk, skin, and oral cavity, in that order, are the primary sources of the microbiome in the gut of piglets immediately after birth. This finding is consistent with the results obtained in the comparison of the microbial compositions of piglets and sows. A previous study has also reported that breast milk plays the most crucial role in the formation of the intestinal microbiome of piglets immediately after birth, with its contribution decreasing over time [62]. In addition, a previous study reported that the composition of the intestinal microbiome of newborns and the composition of maternal breast milk and skin microbiome were similar due to breastfeeding [63]. In the present study, the intestinal microbiota of piglets transferred from maternal sources from birth to 7 days of age were predominantly composed of *Firmicutes* and *Bacteroidota*. These microbes are mainly present in maternal milk and are established in the intestines of piglets through breastfeeding, forming the core microbial clusters early in life [64]. *Firmicutes* can ferment complex carbohydrates that are difficult to break down, resulting in the production of butyrate, a short-chain fatty acid that serves as an energy source for the host. *Firmicutes* also play a role in alleviating intestinal inflammation [65,66]. *Bacteroidota* abundantly produce propionate, a precursor in glucose, lipid, and protein synthesis. This metabolic activity is believed to contribute to improved fetal growth and plays a role in mitigating glucose and lipid disorders in metabolic diseases [66,67]. We confirmed that the microbiome was transferred from sows to piglets and was retained until the post-weaning period. The microbiome transferred from sows to piglets comprises significant components of the piglet intestinal microbiome, and these microbial communities shared between sows and piglets are likely to interact and contribute to the health of both [68,69]. Moreover, microbiomes transferred from the maternal intestines to the piglet intestines after birth are likely to establish early colonization and become significant constituents of the piglet intestinal microbiome. Our results support the theory that the neonatal intestinal microbiome is vertically transmitted from the maternal intestine. Although our study focused on piglets and their relationship with sows, it is essential to consider the influence of the pig farm environment. Additional studies integrating factors related to sows, piglets, and their environments are necessary to better understand the early establishment of the intestinal microbiome in piglets.

## 5. Conclusions

Our findings suggest that the initial composition of the intestinal microbiome is important because bacteria derived from sows that colonize the gastrointestinal tract of piglets maintain a constant ratio despite growth and changes in various environmental factors. This insight could deepen our understanding of the formation of the early microbial community in the intestines of piglets, which receive microbes from sows, and its impact on the health and robustness of piglets. Further research is required to integrate factors related to sows, piglets, and their environments to gain a better understanding of the early establishment of the piglet intestinal microbiome and for the identification of bacterial species.

## Figures and Tables

**Figure 1 animals-13-03378-f001:**
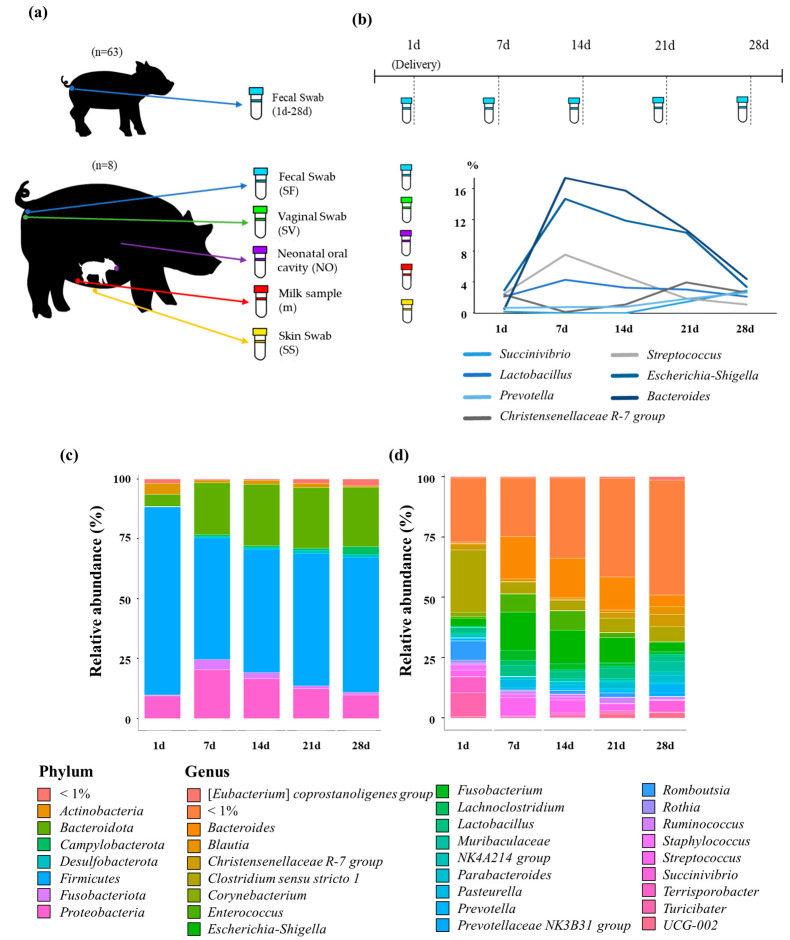
Microbiome sample collection and intestinal microbiome development in piglets 28 d after birth. (**a**) Fecal (SF), skin (SS), milk (m), and vaginal (SV) samples were collected from the sows, and fecal and neonatal oral cavity (NO) samples were collected from the piglets. For piglets, sampling commenced within 24 h of birth and continued up to 28 d. All samples were subjected to 16S rRNA amplicon sequencing of the V3–V4 region. (**b**) Changes in the microbial composition in piglet intestines at specific time points. (**c**,**d**) Stacked bar plots showing the average taxonomic composition of bacteria in piglet feces; (**c**) phylum level; (**d**) genus level (relative abundance > 1%).

**Figure 2 animals-13-03378-f002:**
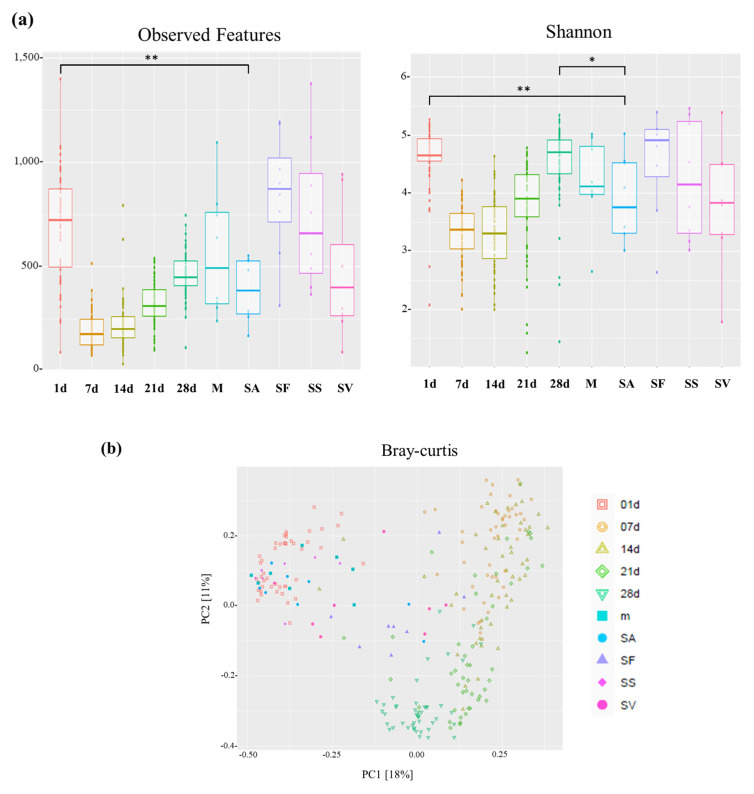
Comparison of bacterial diversity between piglets (*n* = 63) and sows (*n* = 8). (**a**) Boxplots showing the alpha diversity indices (observed features and Shannon diversity, * *p* < 0.05, ** *p* < 0.01). (**b**) Principal coordinate analysis (PCoA) plot comparing the beta diversity of piglet and sow microbiota based on Bray–Curtis distances. m, sow milk; NO, neonatal oral cavity; SF, sow feces; SS, sow skin; SV, sow vagina.

**Figure 3 animals-13-03378-f003:**
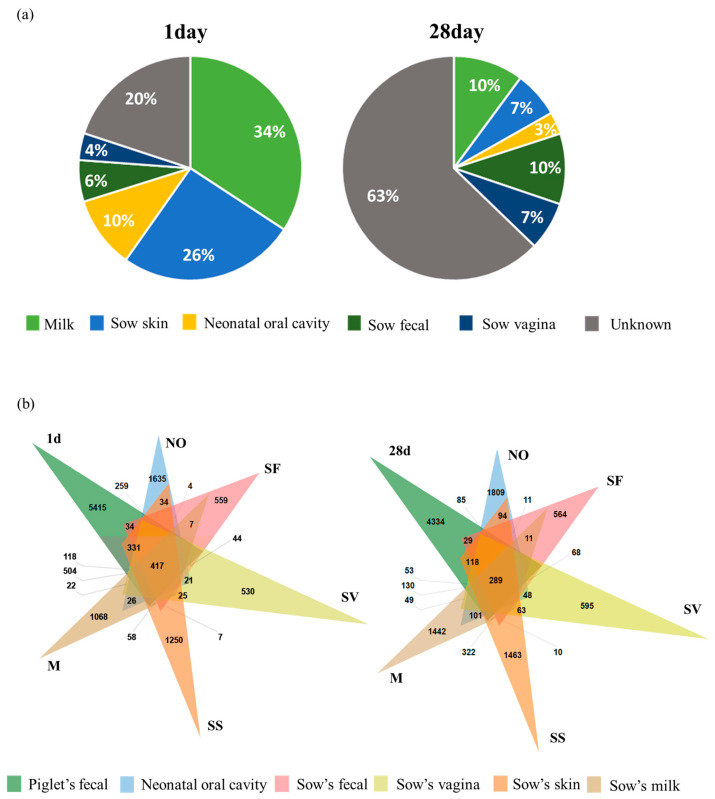
Influence of sow microbiota on the formation of the intestinal microbiome in piglets. (**a**) Contribution of microbial sources in sows to the intestinal microbiome of 1 d and 28 d old piglets. The proportion of “source” from the piglets was estimated from different maternal sources (colored regions) using SourceTracker. (**b**) Number of microbes shared by the sows and piglets. m, sow milk; NO, neonatal oral cavity; SF, sow feces; SS, sow skin; SV, sow vagina.

**Figure 4 animals-13-03378-f004:**
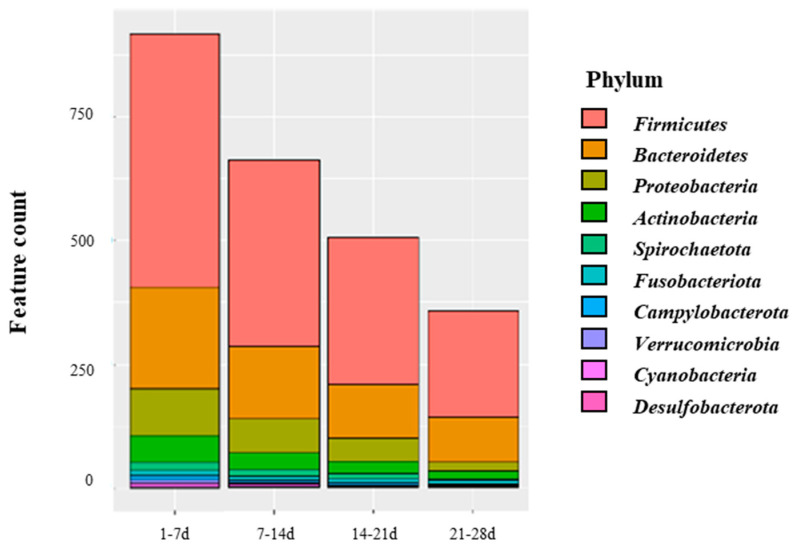
Top 10 bacteria at the phylum level transmitted vertically from sow to piglets at different growth stages.

## Data Availability

The datasets used in this study are available from the corresponding author upon reasonable request.

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
