# Peer review of "Early Colonization of the Intestinal Microbiome of Neonatal Piglets Is Influenced by the Maternal Microbiome"

_animals, 2023, doi:10.3390/ani13213378_

Round 1
Reviewer 1 Report
Comments and Suggestions for Authors
The authors investigated the changes in the fecal microbiome of piglets from birth until weaning, and they were able to present the correlation between the piglet fecal microbiome and sows. They analyzed their results using 16S rRNA sequencing. The introduction was clear and the manuscript had a smooth flow. The results and figures were clearly presented.
One minor comment would be:
In Figure 1 (a) title line 198: It was mentioned that amniotic fluid samples were collected from piglets, it has to be from sows and not piglets.
In this research article, the authors investigated if the piglet intestinal microbiome is vertically transmitted from the mother. The authors compared the fecal microbiome of the piglets to different sample types from the sow mother coming from colostrum, skin, amniotic fluid, and maternal feces.
In fact, the topic is not original because similar studies have been conducted before. For instance, the study by Chen et al. 2022 (Chen, W., Ma, J., Jiang, Y., Deng, L., Lv, N., Gao, J., Cheng, J., Liang, J.B., Wang, Y., Lan, T. and Liao, X., 2022. Selective maternal seeding and rearing environment from birth to weaning shape the developing piglet gut microbiome. Frontiers in Microbiology, 13, p.795101.) was very similar to the study currently presented. They compared the piglet fecal microbiome to different sample types from the sows and they looked at vertical transmission.
Compared to the study by Chen et al., the current study investigated different sample types coming from colostrum which is the pre-milk, and from the amniotic fluid from the sows and correlated them with the fecal microbiome of the piglets. These 2 extra sample types were not reported before.
16S is limited to the genus level; for this reason, it is difficult to investigate the sources of the gut microbiota of piglets at the species level. Metagenomics could be suggested for future work which would add further information to the manuscript which has not been reported before.
The conclusions address the arguments and the research question addressed
References are appropriate, but an important reference to consider in this study the the paper by Chen et al. 2022
Figures and tables represent the results in an efficient way. A minor correction of the title for figure 1 (a) line 198 in which the amniotic fluid was collected from the sow and not from the piglet.
Reviewer 2 Report
Comments and Suggestions for Authors
The manuscript of Lim et al. faces a relevant topic regarding the colonization and establishment of gut microbiota in piglets during nursing.
However, there are lacks in the manuscript. The introduction is short, and it could be improved and extended. The structure and information of the M&M section need to be revised information is missing. Likewise, the results section should be revised.
Below are some comments and suggestions for the authors.
Simple Summary and Abstract report an overall description of the study.
Keywords are missing, please fill this section with appropriate keywords.
Introduction section
This section can be extended and improved.
Line 38. Please, it is not needed to abbreviate SCFAs because the abbreviation is not reported in the text. Then, remove the abbreviation.
M&M section
The ethics statement should be moved to the specific section at the end of the manuscript.
Paragraph 2.2. This section should be named “Animal Management and Experimental Design”. In this section, the authors should include information regarding the housing, e.g. environmental temperature, relative humidity and so on. In addition, the authors stated in the abstract that weaned piglets were fed with solid feed. Therefore, they should add information regarding the feeding regime (e.g., type of feeding and chemical and nutritional composition).
Regarding the “Sample collection”, this section can be integrated into the “Microbial DNA extraction and 16S rRNA gene sequencing” paragraph.
Lines 82 to 85. The authors state: "Fecal, skin, amniotic fluid, and vaginal samples were collected rectally from each animal using a sterile tube??? bio microbial transport medium…".
Can they explain how collected skin, amniotic fluid and vaginal samples from rectal? The entire sentence must be rewritten; indicating that just faecal samples were collected from the rectal ampoule of each animal. Then, the authors must explain each procedure involved for skin, amniotic fluid and vaginal samples collection, respectively. Sample collection methods must be entirely reformulated. In addition, in the results section, the bacterial composition of milk samples is not previously mentioned in the M&M section.
Line 87. There is confusion in describing the sample collection methods. The authors state: “Microbial DNA was extracted from the samples to analyze the microbial communities in the sows and piglets”. Then, the question is: from which samples? To perform a microbial investigation of the gut microbiota (faecal microbiota because the authors stated it in the introduction, line 64) I suppose that the authors used faecal samples. Then, the author must specify what type of sample they used. First of all, faecal samples were used for the characterization of the faecal microbiota and then other samples to evaluate the bacterial composition that piglets vertically received from the sows.
The “Microbial data analysis” section should be numbered as Paragraph 2.3.
Line 110. Please, move the link to the bibliography and then report it as a citation in the manuscript.
Line 115. Please insert the following citation for Bray-Curtis distance.
Bray, J.R.; Curtis, J.T. An Ordination of the Upland Forest Communities of Southern Wisconsin. Ecol. Monogr. 1957, 27, 325–349.
Line 118. Please insert the following citation for the Wilcoxon test.
Wilcoxon, F. Individual comparisons by ranking methods. Biometrics 1945, 1, 80–83.
Results section
The exposition of paragraph 3.1 is confusing. Please, rewrite.
Line 125 to 129. This part can be moved to the M&M section (in paragraph 2.2) because it describes the sample collection methodology.
Figure 2. Please correct the caption in the figure for Bray-Curtis.
Figure 4. It is difficult to comprehend the bar plot of Figure 4. It would be better if the authors could report just the top taxa, such as the first ten major phyla.
I appreciate it if the authors could include in the bibliography of their manuscript the following studies (e.g., underlying the impact of animal nutrition for improving gut microbiota, animal wellness, and growth performance).
Sutera, A.M.; Arfuso, F.; Tardiolo, G.; Riggio, V.; Fazio, F.; Aiese Cigliano, R.; Paytuví, A.; Piccione, G.; Zumbo, A. Effect of a Co-Feed Liquid Whey-Integrated Diet on Crossbred Pigs’ Fecal Microbiota. Animals 2023, 13, 1750. https://doi.org/10.3390/ani13111750
Tardiolo, G.; Romeo, O.; Zumbo, A.; Di Marsico, M.; Sutera, A.M.; Cigliano, R.A.; Paytuví, A.; D’Alessandro, E. Characterization of the Nero Siciliano Pig Fecal Microbiota after a Liquid Whey-Supplemented Diet. Animals 2023, 13, 642. https://doi.org/10.3390/ani13040642
Comments on the Quality of English LanguageMinor editing is required
Reviewer 3 Report
Comments and Suggestions for Authors
The authors a complimented with a well designed and executed study and a well-written manuscript. There is one point that needs some clarification: in the entire paper I could not find the time point of weaning (or did I miss something?). Does this mean that day 28 is considered as a common weaning time. This would imply that no samples from weaned piglets are included in this study. However, in some place and in line 247 also it is states that the weaning period is between day 14 to day 28. Does this mean that during this period piglets received already soli feed while still staying with the sow. Pleas clarify (at least in M&M)
Minor editorials
Line 21: Consider replacing 16D gene by 16S genome
Line 82: delete or replace the word “rectally” (see section 3.1.).
Line 177: piglets shared OTUs with (replace “in” by “with” – 4 time in this sentence)
Line 261: you state that the mother’s microbiome is partly transferred through the placenta. Pleas be more precise as it is likely that you refer here to microbial metabolites that are transferred through the placenta (in contrast to amnion fluid which indeed might be colonized by maternal microbiota, see line 272-273).
Round 2
Reviewer 2 Report
Comments and Suggestions for Authors
I thank the authors for following my comments and suggestions.
The manuscript is improved and just needs to be formatted according to the Animals MDPI template, also improving the quality of the figures.
The names of phyla, families and genera must be reported in the manuscript in Italics style (e.g., see lines 272, 280, and so on).
I recommend the manuscript for publication. However, just a few issues need to be fixed in the text before publication.
Lines 56-58. “In particular, breast milk has been confirmed to play a role in increasing beneficial bacteria and reducing harmful bacteria in the intestinal microbiome [20,21]”.
I thank the authors for including the two suggested studies in the introduction of their manuscript. However, the sentence needs to be revised. In the suggested studies the authors supplemented growing piglets with fresh liquid whey and not with breast milk. Therefore, I suggest revising the sentence as follows: “In particular, the supplementation of fresh liquid whey in growing piglets has been confirmed to play a role in increasing beneficial bacteria and reducing harmful bacteria in the pig faecal microbiota [20,21]”.
Line 83. Please, state “ad libitum” and “ad libitum-fed” in Italics style.
Figure 4. The figure is improved, and the distinction among the phyla is better than the previous barplot. However, there is an error in the caption “Txon”. I suggest replacing the caption with “Phylum”.
Comments on the Quality of English LanguageMinor editing of English language required.
